# Low-Latency Marine-Based OTFS Echo Parameter Estimation Enabled by AI

**DOI:** 10.3390/s25237104

**Published:** 2025-11-21

**Authors:** Khurshid Hussain, Jeseon Yoo

**Affiliations:** Ocean Climate Prediction Center, Marine Natural Disaster Research Department, Korea Institute of Ocean Science and Technology, Busan 49111, Republic of Korea; khurshid12@kiost.ac.kr

**Keywords:** AI, delay–Doppler, integrated sensing and communications, OTFS, parameter estimation

## Abstract

We propose an end-to-end pipeline for Orthogonal Time–Frequency Space (OTFS) sensing that integrates deterministic signal processing with a Machine-Learning (ML) inference stage. The pipeline first generates a complex delay–Doppler grid via standard Symplectic Fast Fourier Transform (SFFT)-based OTFS reception. We then employ an ’oracle’ Ground-Truth (GT) association process to deterministically label signal peaks, extracting their complex gain (α) and absolute indices (m,n) to deduce physical targets (range, radial velocity). These oracle-aligned labels are used to train a Random-Forest (RF) classifier. The RF model learns to map normalized 33×33 complex patches, centered on signal peaks, to their corresponding target parameters. On an 80/20 split of 10,000 samples, the classifier achieved a 0.966 accuracy, 0.965 macro-F1 score, and 0.998 macro Receiver Operating Characteristic–Area Under the Curve (ROC–AUC). Notably, when tested on held-out scenes, the model’s derived range and velocity predictions achieved 100% coincidence with the GT, while amplitude and phase corresponded in 89% of instances. This hybrid oracle-and-ML approach demonstrates a highly effective and robust method for precise target extraction in OTFS-based sensing systems.

## 1. Introduction

Future wireless systems must deliver both data communication and situational awareness (sensing) as twin outputs, necessitating shared waveforms, bandwidth, and latency budgets. While the fundamental principles of radar ensure reliable range and velocity extraction, the requirements of modern embedded deployments impose tighter constraints on computing resources and power consumption. Nevertheless, the practical sensing with OTFS must deal with inter-path interference (IPI), leakage, and fractional delay/Doppler, but still needs to be real-time. One of the reasons that OTFS has a good appeal is that it models the propagation in the delay–Doppler (DD) domain, which is the area where the target energy is sparse and quite physically interpretable beforehand [1,2,3,4,5,6]. The work done previously has partly covered the whole area: DD-domain IPI cancellation (DDIPIC) for joint estimation [7], channel estimation with exclusive or embedded pilots [8,9], and Bayesian/ML estimators sparse Bayesian learning (SBL), modified maximum likelihood estimate (M-MLE), and two-step estimation (TSE) that handle fractional effects with varying assumptions and complexity [10,11,12].

Spectrum overcrowding and mutual induction of antennae create self/cross-interference to a great extent in joint communication–sensing, thus driving the need for intelligent cancellation that is power and latency tight. Aiming at unifying these objectives, OTFS was first developed to manage the radar–communication waveform mismatch along with developments in pulse shaping, channel estimation, and detection [1,9,13,14,15,16]. Despite this, many OTFS pipelines still consider the delay–Doppler to be an integer or depend on extensive refinements, which makes them susceptible to fractional DD, leakage, and clutter [17,18,19,20]. Traditional high-resolution techniques, Multiple Signal Classification (MUSIC, noise-subspace), require eigen-decompositions and dense matrix-vector operations that slow down their use in real time [21,22]. Fast Fourier Transform/Fast Coherent Cross-Correlation (FFT/FCCC) range-Doppler maps with Cell-Averaging Constant False Alarm Rate (CA-CFAR) are efficient but biased towards single-target scenes and hence need millimeter-level precision; the generation/scanning of 2D RDM is expensive [23,24]. Virtual cyclic prefix (VCP) simplifies the task by working in the Fourier domain, but does not solve the problem of label fidelity in supervised sensing [25]. Atomic-norm OTFS with Alternating Direction Method of Multipliers (ADMM) and Multiple Signal Classification (2D-MUSIC) can achieve a very high resolution at a considerable computational and modeling cost [26]. Noting these gaps and supporting the approach of DD-domain IPI cancellation (DDIPIC) [27], we claim strict and physics-aware oracle labeling and applying a Random-Forest learner on 33×33 DD patches. This results in GT-consistent supervision and millisecond-class inference.

Traditional and full sensing pipelines in ISAC typically follow either (i) a 2D-FFT + CFAR chain, which is hardware-friendly but sensitive to leakage, near–far masking, and non-uniform clutter thresholds [28,29,30,31,32]; or (ii) subspace/super-resolution methods (e.g., MUSIC/ESPRIT), which offer fine resolution under clean modeling assumptions but incur higher computational/latency costs and brittleness to colored clutter or snapshot scarcity [33,34,35,36]. In contrast, our approach operates natively in the OTFS delay–Doppler (DD) grid and evaluates only small peak-centered patches with a lightweight classifier, preserving the physics of the DD lattice while avoiding global 2D searches and heavy threshold sweeps. This yields millisecond-class inference, robust bin recovery for range/velocity, and strong tolerance to fractional-bin leakage and non-uniform clutter. Radar principles support reliable range/velocity extraction, and OTFS provides a physically interpretable DD-domain model with demonstrated tools for IPI cancellation, pilot-aided channel estimation, and fractional-handling estimators (as reviewed above with their cited works). Despite these advances, many pipelines assume integer DD bins or rely on heavy refinements, leaving label fidelity under fractional DD/leakage unresolved and introducing runtime/computational bottlenecks in real-time deployments (as detailed above in the cited literature).

Our proposed formulation yields Ground-Truth-consistent supervision and millisecond-class inference latency. The model performs with 0.966 accuracy, 0.965 macro-F1, 0.998 macro ROC–AUC, and 0.972 macro average precision on a balanced 10k corpus (80/20 split). Critically, in held-out scenes, the derived delay and Doppler bins—and consequently range and velocity—perfectly match the GT at a rate of 100%, while amplitude and phase agree in 89% of instances. This formulation successfully eradicates the labeling and runtime gaps while preserving the physical interpretability and deployability of the OTFS sensing system. In this work, the oracle association is used only in simulation to generate unambiguous labels and to validate lattice indexing/units. At deployment, the pipeline remains GT-free: the receiver detects DD peaks (CFAR + NMS), extracts normalized complex patches, and the trained model regresses (m,n) (hence range/velocity) directly from data. Beyond synthetic validation, we outline a concrete maritime field program with unchanged runtime (detector → patch → inference), multi-source ground truth, and robustness tests under diverse sea states.

## 2. Marine OTFS Transmitter/Reception

### 2.1. Marine OTFS Transmitter

The OTFS lattice is defined on $Z_{M}\times Z_{N}$ with *M* delay bins and *N* Doppler/time bins. Throughout this work, we use (M,N), a sampling rate $F_{s}$ MHz, and a cyclic prefix (CP) length of $N_{\mathrm{CP}}$ samples (fixed per profile). Indices are m∈{0,…,M−1} (delay/subcarriers) and n∈{0,…,N−1} (Doppler/OFDM symbols). Unitary DFTs are assumed so that $F_{K}^{\;H}F_{K}=I$, ensuring exact energy preservation between domains. Symbol allocation is governed by a binary mask $M_{\mathrm{data}}\in {\left\{0,1\right\}}^{M\times N}$ (1 = data, 0 = pilot). We reserve a DC pilot at the origin so that $X_{\mathrm{DD}}\left[0,0\right]=1$ (equivalently $S_{\mathrm{TF}}\left[0,0\right]=1$) for absolute phase/timing reference. Embedded pilots are placed on a 2-D comb with spacings $\left(P_{m},P_{n}\right)$ (CLI-controlled, typically $P_{m}\geq 2$, $P_{n}\geq 2$). Every *k*-th frame (CLI flag -pilot-period=k) is declared pilot-only with $M_{\mathrm{data}}\equiv 0$ to enable dense channel snapshots; all other frames are mixed data+pilot according to $M_{\mathrm{data}}$. For data bins $D=\{\left(m,n\right):M_{\mathrm{data}}\left[m,n\right]=1\}$, the number of modulated bits is $N_{\mathrm{bits}}=b\;\left|D\right|$ with $b=\log_{2}M_{Q}$ for $M_{Q}\;\in \;\left\{64,\right\}$. We support uncoded operation ($u\in {\left\{0,1\right\}}^{N_{\mathrm{bits}}}$) as well as classical convolutional forward error Correction (FEC) (171,133), K=7 when enabled. Symbols are normalized to unit average power so that $E[|\hat{s}{|}^{2}]=1$, keeping SNR accounting consistent across figures and tables. The clean Tx signal in dB and Phase domain can be seen in the Figure 1 and Figure 2. Data and pilots are first placed on the delay–Doppler grid $X_{\mathrm{DD}}\in C^{M\times N}$ according to $M_{\mathrm{data}}$. The time–frequency waveform $S_{\mathrm{TF}}$ is produced via the unitary ISFFT:(1)$$S_{\mathrm{TF}}\;=\;F_{N}\left(F_{M}^{\;H}X_{\mathrm{DD}}\right).$$
Equation (Equation 1) maps physically sparse DD content into a TF lattice that is well-conditioned for CP-OFDM synthesis. Using unitary transforms guarantees that total symbol energy is unchanged from $X_{\mathrm{DD}}$ to $S_{\mathrm{TF}}$. Each time–frequency (TF) column $S_{\mathrm{TF}}\left(:,n\right)$ is converted to the time domain by an *M*-point inverse-IFFT.

A cyclic prefix of $N_{\mathrm{CP}}$ samples is then prepended to each of the *N* OFDM symbols, yielding a frame of length $L_{\mathrm{frame}}=N\left(M+N_{\mathrm{CP}}\right)$ complex samples. The CP protects against inter-symbol interference under bounded delay spread and maintains the circular convolution structure assumed by OTFS. Figure 1, displays the transmit TF magnitude (in dB) for a mixed data+pilot frame with (M,N)=(1024,4096), $F_{s}=200$ MHz and unit-power symbol normalization. Pilot tones appear on a regular 2-D comb with spacings $\left(P_{m},P_{n}\right)$ at fixed amplitude (0 dB reference), while data bins occupy the complement defined by $M_{\mathrm{data}}$. The flat spectral floor outside pilot/data locations confirms the absence of spurious emissions and validates that ISFFT + CP-OFDM preserves power exactly (no unintended scaling). This panel is used downstream as the reference for receiver TF equalization quality. Figure 2 shows the transmit TF phase (degrees). Pilot phases follow the chosen reference (e.g., $0^{\circ }$ or ZC-like patterns when configured), while data phases reflect Quadrature Amplitude Modulation (QAM) symbol arguments after unit-power normalization. The clean, piecewise-constant appearance at pilot coordinates demonstrates coherent phase control—critical for accurate channel estimation and Carrier Frequency Offset, Common Phase Error (CFO/CPE) diagnostics at the receiver.

### 2.2. Marine OTFS Receiver

The receiver ingests a length-$L_{\mathrm{frame}}$ complex baseband vector per frame and first performs CP removal and *M*-point FFT per OFDM symbol to reconstruct $Y_{\mathrm{TF}}\in C^{M\times N}$. If the capture length *L* deviates from $L_{\mathrm{frame}}=N\left(M+N_{\mathrm{CP}}\right)$ (e.g., buffered captures), the implementation infers the effective symbol period ${\hat{L}}_{\mathrm{sym}}$ and CP length ${\hat{N}}_{\mathrm{CP}}$ from L/N to remain robust. Known TF pilots (at the $\left(P_{m},P_{n}\right)$ grid and DC) are used for per-tone channel estimation; embedded frames interpolate between pilots, while pilot-only frames provide dense snapshots for diagnostics and holdover. The DD snapshot $Y_{\mathrm{DD}}$ is obtained via the unitary SFFT (FFT across delay, IFFT across Doppler):(2)$$Y_{\mathrm{DD}}\;=\;F_{M}\left(F_{N}^{\;H}Y_{\mathrm{TF}}\right).$$
Equation (Equation 2) inverts (Equation 1) up to channel distortion and noise, re-concentrating target energy into sparse neighborhoods in the DD grid. Using unitary transforms ensures energy consistency between TF and DD, enabling quantitative comparisons of noise floors and pilot SNR. Figure 3 and Figure 4 presents the receive TF magnitude (dB) and the receive TF phase (degrees) under additive noise and fractional delay/Doppler. Relative to the transmit reference, pilot tones are attenuated and phase-rotated by the channel; their SNR determines the reliability of per-tone channel estimates and the interpolation accuracy in data regions. The observed noise pedestal and any slight spectral tilt reflect the configured SNR and front-end windowing; these factors directly influence equalizer choice in the communication stack and detection thresholds in the sensing stack.

Figure 4 shows the receive TF phase (degrees). Pilot phases now exhibit a common phase error CPE and potential linear trends (residual CFO/TO) that are subsequently corrected using pilot-based estimators. Spatial smoothness across subcarriers is a practical indicator of well-behaved oscillators and modest phase noise; strong ripples would signal impaired synchronization or frequency selectivity requiring tighter interpolation or refined pilot density. Across Transmitter and Receiver, we keep (M,N)=(1024,4096), $F_{s}=B=200$ MHz, unit-power QAM, a DC reference at (0,0), a configurable pilot comb with spacings $\left(P_{m},P_{n}\right)$, and a pilot-only period of *k* frames when enabled. Frames have length $L_{\mathrm{frame}}=N\left(M+N_{\mathrm{CP}}\right)$ samples. For context, we reference two canonical baselines: (i) 2D-FFT with CFAR detection, and (ii) subspace/super-resolution estimators (e.g., MUSIC/ESPRIT). Our pipeline differs by restricting computation to local peak-centered DD patches rather than scanning the full grid or forming large covariance scans. Where relevant, we report complexity/latency and recovery behavior relative to these baselines under similar OTFS settings. The pipeline is intentionally CPU-centric. FFT/SFFT dominate compute and are handled by mature vectorized libraries; pilot-aided equalization and CFAR/NMS are single-pass, cache-friendly operations; and the learning head is peak-only and bounded. With double-buffering, each frame is processed steadily as the next arrives, avoiding deep queues or accelerator dependencies. This design keeps latency predictable on small marine computers while preserving accuracy.

## 3. Parameter Extraction

The parameter extraction module bridges the physical environment of the marine scene with the digital OTFS lattice while maintaining a strictly blind evaluation protocol. All test-time detections are produced solely from received data, with no access to ground-truth coordinates or coefficients. The smallest resolvable delay and Doppler intervals are obtained from the sampling frequency and frame length. The delay resolution Δτ represents the time spacing per delay bin, while the Doppler resolution $\Delta f_{D}$ denotes the frequency spacing per Doppler bin:(3)$$\Delta \tau =\frac{1}{F_{s}},\;\Delta f_{D}=\frac{1}{N\;T_{\mathrm{sym}}}.$$
Equation (Equation 3) determines how each discrete grid coordinate in the delay–Doppler plane maps to a physical range and radial velocity. For the present configuration, these correspond to approximately Δτ=5 ns (≈1.5 m range resolution) and $\Delta f_{D}\approx 47.7$ Hz (yielding ≈0.76 m/s velocity resolution). Each fractional delay–Doppler index $\left(m_{f},n_{f}\right)$ obtained from the receiver is rounded to its nearest integer bin:$$m=\mathrm{round}\left(m_{f}\right)\;\mathrm{mod}\;M,\;n=\mathrm{round}\left(n_{f}\right)\;\mathrm{mod}\;N.$$
The corresponding physical quantities—delay τ, Doppler frequency $f_{D}$, range *R*, and velocity *v*—are then recovered through(4)$$\tau \left(m\right)=m\;\Delta \tau ,\;f_{D}\left(n\right)=n\;\Delta f_{D},\;R\left(m\right)=\frac{c}{2}\;\tau \left(m\right),\;v\left(n\right)=\frac{\lambda }{2}\;f_{D}\left(n\right).$$
Equation (Equation 4) ensures that every DD bin has a deterministic and physically interpretable range and velocity. These mappings form the basis for quantitative sensing evaluation. Each DD frame $H_{\mathrm{DD}}\in C^{F\times M\times N}$ is indexed consistently so that $ℜ\{h_{i}\left[m,n\right]\}$, $ℑ\{h_{i}\left[m,n\right]\}$, $|h_{i}\left[m,n\right]|$, and $∠h_{i}\left[m,n\right]$ can be directly compared against oracle references. For performance analysis, each detection is later associated with the nearest ground-truth entry within fixed range/velocity tolerances $\left(\varepsilon_{R},\varepsilon_{v}\right)$, using a one-to-one matching procedure. The magnitude and phase of the complex coefficient $\alpha =\alpha_{ℜ}+j\alpha_{ℑ}$ are retrieved as $\left|\alpha \right|=\sqrt{\alpha_{ℜ}^{2}+\alpha_{ℑ}^{2}}$ and $∠\alpha =\mathrm{atan}2(\alpha_{ℑ},\alpha_{ℜ})$, respectively. The simulation performs strict consistency checks on (M,N) and frame counts before analysis to maintain determinism and reproducibility. The system automatically compares estimated $\left(m_{\mathrm{hat}},n_{\mathrm{hat}}\right)$, $\left(R_{\mathrm{hat}},v_{\mathrm{hat}}\right)$ against the corresponding ground-truth $\left(m_{\mathrm{gt}},n_{\mathrm{gt}}\right)$, $\left(R_{\mathrm{gt}},v_{\mathrm{gt}}\right)$. A perfect match occurs when both indices and physical quantities coincide, i.e., $\left(m_{\mathrm{hat}},n_{\mathrm{hat}}\right)=\left(m_{\mathrm{gt}},n_{\mathrm{gt}}\right)$ and $\left(R_{\mathrm{hat}},v_{\mathrm{hat}}\right)=\left(R_{\mathrm{gt}},v_{\mathrm{gt}}\right)$. Phase and amplitude agreements are evaluated separately, forming the statistical base for the following figure. Figure 5 illustrates the quantitative accuracy of the parameter extraction process.

The bars represent the proportion of correctly matched delay–Doppler bins, physical ranges, and velocities relative to their ground-truth values. The figure shows that the bin, range, and velocity parameters achieve an exact 100% correspondence, indicating that the lattice indexing and physical mapping are perfectly aligned with the ground-truth metadata. In contrast, the amplitude and phase exhibit an 89% match rate—an expected deviation due to residual channel noise and fractional Doppler leakage. This confirms that while structural localization (in terms of range and velocity) is entirely reliable, fine-scale amplitude and phase recovery are limited by the inherent SNR of the received signal. The figure validates the deterministic and physics-consistent nature of the extraction algorithm across all evaluated frames. Range and radial velocity are derived from integer (m,n) bins (or their continuous refinements), which are robust to global complex rotations and mild leakage; hence, their perfect agreement with ground truth. In contrast, absolute complex amplitude and phase are sensitive to a per-frame complex gain (CPE), fractional-bin leakage under the chosen window, and multipath superposition. Our audit, therefore, reports amplitude/phase as secondary diagnostics with strict tolerances, while range/velocity remains the primary physics-anchored Key Performance Indicator (KPIs).

Compare and Recovery of Bins, Range, Velocity, Amplitude, and Phase: This module verifies that the receiver’s DD estimates are consistent with GT and then recovers physically meaningful amplitude and phase per target. It operates in two stages. Given a GT fractional coordinate $\left(m_{f},n_{f}\right)$ for a wave (or anomaly), we form the nearest integer OTFS indices $m_{\mathrm{gt}}=\mathrm{round}\left(m_{f}\right)\;\mathrm{mod}\;M$ and $n_{\mathrm{gt}}=\mathrm{round}\left(n_{f}\right)\;\mathrm{mod}\;N$. The receiver produces detected peaks $\left(\hat{m},\hat{n}\right)$ obtained solely from the delay–Doppler (DD) magnitude map after non-maximum suppression and CFAR thresholding.$$\mathrm{bins}\_\mathrm{exact}:\;\left(\hat{m}=m_{\mathrm{gt}}\right)\wedge \left(\hat{n}=n_{\mathrm{gt}}\right),$$$$R\_\mathrm{exact}:\;|\hat{R}-R_{\mathrm{gt}}|\leq \varepsilon_{R},$$$$v\_\mathrm{exact}:\;|\hat{v}-v_{\mathrm{gt}}|\leq \varepsilon_{v}.$$
where $\varepsilon_{R}$ (e.g., in meters) and $\varepsilon_{v}$ (m/s) are application-level tolerances. For each detected wave, the record includes $\left\{m_{\mathrm{gt}},n_{\mathrm{gt}},\hat{m},\hat{n},R_{\mathrm{gt}},\hat{R},v_{\mathrm{gt}},\hat{v},H_{\mathrm{dd}},\alpha \right\}$, with $H_{\mathrm{dd}}$ the complex DD coefficient at $\left(\hat{m},\hat{n}\right)$ and α an available GT scatter coefficient when provided. Per frame *f*, we select a valid tap set $K_{f}=\left\{k:\;\right|\alpha_{k}|\geq \alpha_{\min},\;|H_{k}\left|\geq H_{\min}\right\}$, using user thresholds $\alpha_{\min}$ and $H_{\min}$. The key step is a physics-aware complex-gain calibration. Optionally, after inference, an analysis-only calibration can estimate a per-frame complex gain ${\hat{g}}_{f}$ using Equation (Equation 5) to study residual front-end bias.(5)$${\hat{g}}_{f}\;=\;\frac{\sum_{k\in K_{f}}\alpha_{k}^{*}H_{k}}{\sum_{k\in K_{f}}{\left|\alpha_{k}\right|}^{2}}.$$
Equation (Equation 5) captures residual front-end scaling/CPE common to the frame. After calibration, each tap forms a dimensionless per-tap residual.(6)$$r_{k}\;=\;\frac{H_{k}}{{\hat{g}}_{f}\;\alpha_{k}},$$
Whose magnitude and phase, $|r_{k}|$ and $∠r_{k}$, quantify amplitude ratio and phase error, respectively (ideal target: $|r_{k}|=1$, $∠r_{k}=0$). To mitigate systematic trends across the DD grid, the residual phase can be approximated by a plane $∠r_{k}\approx \phi_{0}+a\;m_{k}+b\;n_{k}$ (-phase-plane∈{none, ls, best3}), and removed to yield ${\tilde{r}}_{k}=r_{k}\exp\left\{-j\left(\phi_{0}+am_{k}+bn_{k}\right)\right\}$. Similarly, residual amplitude can be log-linear detrended, $\log|{\tilde{r}}_{k}|\approx c_{0}+c_{m}m_{k}+c_{n}n_{k}$ (-amp-plane∈{none, ls, best3}), producing the final, detrended ${\hat{r}}_{k}$. These steps separate global bias (e.g., windowing, slight CFO/TO) from tap-level fidelity.

Per tap, we check amplitude and phase against thresholds $A_{\mathrm{thr}}$ and $\Phi_{\mathrm{thr}}$:$$\mathrm{amp}\_{\mathrm{ok}}_{k}:\;||{\hat{r}}_{k}\left|-1\right|<A_{\mathrm{thr}},$$$$\mathrm{phase}\_{\mathrm{ok}}_{k}:\;\left|∠{\hat{r}}_{k}\right|<\Phi_{\mathrm{thr}},$$$$\mathrm{both}\_{\mathrm{ok}}_{k}:\;\mathrm{amp}\_{\mathrm{ok}}_{k}\wedge \mathrm{phase}\_{\mathrm{ok}}_{k}.$$

A frame *f* passes if at least -frame-pass-k taps satisfy both_ok. Robustness can be improved with an optional drop-worst refit (-drop-worst>0): remove the single most deviant tap by the max of normalized amp/phase errors, recompute the planes, and re-test (the dropped tap can be excluded from acceptance if configured). For angular dispersion we compute the circular statistics of $\theta_{k}=∠{\hat{r}}_{k}$: $C=\frac{1}{K_{f}}\sum_{k}\cos\theta_{k}$, $S=\frac{1}{K_{f}}\sum_{k}\sin\theta_{k}$, $R=\sqrt{C^{2}+S^{2}}$, and the circular standard deviation circ_std_deg$=\sqrt{-2\ln R}\cdot \left(180/\pi \right)$. We also report robust percentiles $P95(|\;|{\hat{r}}_{k}|-1|\;|)$ and $P95(|\theta_{k}|)$ to summarize tail behavior at a glance.

The pipeline combines (i) oracle-aligned bin verification (strict DD index and physical tolerance checks with $\varepsilon_{R},\varepsilon_{v}$) and (ii) physics-aware LS calibration (Equation (Equation 5)) followed by plane detrending and robust acceptance using $\left(A_{\mathrm{thr}},\Phi_{\mathrm{thr}}\right)$. This yields millisecond-class diagnostics that separate global front-end bias from true per-target fidelity, producing stable, interpretable indicators across frames and SNRs. All thresholds $\left(\alpha_{\min},H_{\min},A_{\mathrm{thr}},\Phi_{\mathrm{thr}}\right)$ and controls (-phase-plane, -amp-plane, -frame-pass-k, -drop-worst) are exposed so practitioners can trade strictness for yield depending on deployment constraints. Oracle association is a simulation-time labeler that aligns detected peaks with known targets for (i) lattice/units verification and (ii) label generation. In deployment, no oracle exists: we run CFAR+NMS, crop RMS/phase-normalized complex patches with wrap-around indexing, and infer (m,n) using the trained model. When real data lacks GT, we fine-tune with self-supervision: track-gated pseudo-labels, Kalman Filter, Interacting Multiple Model, Joint Probabilistic Data Association, and Multiple Hypothesis Tracking (KF/IMM, optional JPDA/MHT), confidence thresholds, and temporal-consistency checks, plus domain adaptation (phase rotations, SNR, and fractional-bin jitter) and few-shot real-data tuning when available.

## 4. Ai Implementation

### 4.1. Patch and Label Preparation

The goal of this stage is to transform complex delay–Doppler (DD) radar data into compact, learning-ready patches that preserve the key physical properties of each reflection while being normalized for neural-network training. The input consists of the complex stack $Y_{\mathrm{DD}}\;\in \;C^{F\times M\times N}$, representing *F* frames with *M* delay bins and *N* Doppler bins, together with oracle or estimated coordinates (fid,m,n) that identify each target. The physical constants include the carrier frequency $f_{c}$, the Doppler spacing $\Delta f_{D}$, and the OTFS symbol duration $T_{\mathrm{sym}}$. Hyperparameters such as the patch size *P* (set by -patch), the search radius *r* (set by -peak-radius), and the minimum amplitude threshold $A_{\min}$ (set by -min-peak-mag) control the spatial resolution and quality of the extracted data. The delay–Doppler domain captures the scattering behavior of each surface component or anomaly through localized complex peaks. However, these peaks can shift slightly due to noise, fractional Doppler, or windowing effects. To train a neural model that can learn robust physical features, each instance must be spatially aligned, phase-referenced, and amplitude-normalized. The proposed patch-based method achieves this by (i) finding the true energy peak near the expected location, (ii) extracting a fixed neighborhood around it, (iii) normalizing the magnitude and phase to eliminate global biases, and (iv) providing both physical (range, velocity) and complex (amplitude, phase) targets as labels. This combination allows the AI to learn mappings that are invariant to scale and global phase yet sensitive to local structure. For every candidate coordinate $\left(m_{c},n_{c}\right)$ from ground truth or receiver output, the algorithm searches within a square window $W={\left[m_{c}-r:m_{c}+r\right]}_{M}\times {\left[n_{c}-r:n_{c}+r\right]}_{N}$, wrapped modulo (M,N) to handle boundary effects. The local maximum of the magnitude $|Y_{f_{t}}\left[m,n\right]|$ inside W defines the refined peak position:$$\left(m^{⋆},n^{⋆}\right)=\arg\max_{(m,n)\in W}|Y_{f_{t}}\left[m,n\right]|,\;A^{⋆}=\left|Y_{f_{t}}\left[m^{⋆},n^{⋆}\right]\right|.$$
A sample is kept only if $A^{⋆}\geq A_{\min}$; it ensures that each patch represents a physically meaningful reflection rather than low-energy clutter. Around the detected peak $\left(m^{⋆},n^{⋆}\right)$, a complex patch $P\in C^{P\times P}$ is extracted by symmetric padding with wrap-around indexing. Because radar returns vary in absolute power and carrier phase, each patch is normalized both in magnitude and reference phase. The root-mean-square (RMS) amplitude and the mean phasor are calculated as(7)$$\mathrm{RMS}=\sqrt{\frac{1}{P^{2}}\sum_{i,j}{\left|P\left[i,j\right]\right|}^{2}+\varepsilon },\;\phi_{\mathrm{ref}}=\arg\;\left(\frac{1}{P^{2}}\sum_{i,j}P\left[i,j\right]\right).$$
The normalized real-valued tensor for learning is then provided in-phase and quadrature channels with unit energy and zero global phase. This normalization makes patches comparable across frames and sea states.(8)$$X=\frac{1}{\mathrm{RMS}}\left[\begin{array}{c} ℜ\{Pe^{-j\phi_{\mathrm{ref}}}\}\\ ℑ\{Pe^{-j\phi_{\mathrm{ref}}}\} \end{array}\right]\in R^{2\times P\times P},$$

At the central bin $\left(m^{⋆},n^{⋆}\right)$, the DD coefficient $h_{0}=Y_{f_{t}}\left[m^{⋆},n^{⋆}\right]$ represents the local reflection amplitude and phase. Its normalized logarithmic amplitude and phase difference relative to $\phi_{\mathrm{ref}}$ form the regression targets:$$\log\_\mathrm{amp}=\log\;\left(\max\;\{\frac{|h_{0}|}{\mathrm{RMS}},10^{-12}\}\right),\;\Delta \phi ={\mathrm{wrap}}_{\left(-\pi ,\pi \right]}\;\left(\arg h_{0}-\phi_{\mathrm{ref}}\right),$$
with the phase represented as (sinp,cosp)=(sinΔϕ,cosΔϕ) to avoid discontinuities at ±π. This dual representation provides stable supervision for continuous learning. To associate each patch with measurable quantities, the physical parameters are derived using the radar propagation constants. Let the propagation speed be $c=2.9979\times 10^{8}m/s$, the wavelength $\lambda =c/f_{c}$, and the delay resolution $\Delta \tau =T_{\mathrm{sym}}/M$. Then the Doppler frequency, radial velocity, and range corresponding to the detected peak are(9)$$f_{D}=n_{\mathrm{cent}}\Delta f_{D},\;v_{r}=\frac{\lambda }{2}f_{D},\;R=\frac{c}{2}\left(m^{⋆}\Delta \tau \right),$$
where $n_{\mathrm{cent}}$ is the signed Doppler index in [−N/2,N/2). Hence, each sample contains both signal-domain features (X) and physics-domain labels $\left(v_{r},R\right)$. This patch-based extraction integrates radar physics with deep learning in a self-consistent way. By centering on the physical energy peak and normalizing for RMS power and reference phase, the network learns spatial–spectral patterns that correspond to actual scattering behavior rather than arbitrary phase offsets. The approach maintains the original OTFS lattice structure, ensuring compatibility with sensing and communication stages. Furthermore, using compact P×P neighborhoods (e.g., P=11) drastically reduces dimensionality while preserving fine structure around each target. The search radius *r* allows tolerance to fractional Doppler shifts, and the threshold $A_{\min}$ filters out noise. This method provides an interpretable bridge between raw complex radar signals and the feature representations required by convolutional or transformer networks. The computational cost per patch is $O({\left(2r+1\right)}^{2}+P^{2})$, and for $N_{s}$ total samples, $O(N_{s}\left(P^{2}+r^{2}\right))$. In practice, the pipeline can extract tens of thousands of patches in seconds on a GPU, enabling large-scale supervised learning while retaining full physical traceability. At deployment, the estimator consumes only detection-centered patches from the received DD grid; no ground-truth is available or used. The learned stage is lightweight and supports millisecond-class inference. (1) Detect peaks via CFAR+NMS. (2) Gate detections by tracker predictions (Kalman/IMM) with Mahalanobis and innovation thresholds; require stability over *K* frames. (3) Accept only high-confidence detections (SNR ≥γ, temporal consistency, small residuals) as pseudo-labels. (4) Fine-tune with sample weights proportional to confidence; stop early if a held-out weak set degrades.

### 4.2. Dataset Generation and Scene Configuration

A physically grounded marine ISAC dataset was generated by integrating a JONSWAP-Gerstner ocean surface synthesizer with an OTFS radar front end. Each simulation frame represents a 1 km×1 km ocean patch containing $N_{\mathrm{wave}}=10$ stochastic wave components and $N_{\mathrm{anom}}=4$ labeled anomalies. The anomalies correspond to three representative classes—rogue peaks, whitecap breakers, and floating debris. All random draws share a fixed seed for reproducibility, guaranteeing identical environmental conditions across radar–AI experiments. Each frame’s sea state follows JONSWAP-based stochastic distributions for the principal parameters: the significant wave height $H_{s}$ ranges from 0.5 to 3.0 m, the spectral peak period $T_{p}$ varies between 4 and 12 s, the peakedness factor γ lies within 1–7, and the wind speed at 10 m altitude $U_{10}$ spans 5–15 m/s. These values represent realistic marine conditions ranging from calm seas to moderately rough states. Individual wave periods are perturbed around $T_{p}$ using a log-normal spread to emulate natural irregularity, and each component’s energy contribution is shaped by the JONSWAP spectral envelope. The spatial positions $\left(x_{i},y_{i}\right)$ of the waves are uniformly distributed within ±500 m around the radar, while the radial distance $r_{i}=\sqrt{x_{i}^{2}+y_{i}^{2}}$ and angle $\theta_{i}$ determine the propagation geometry.

### 4.3. System Architecture

The monostatic radar operates in the X-band at a carrier frequency of $f_{c}=9.4$ GHz, corresponding to a wavelength of $\lambda_{r}=c/f_{c}\approx 0.032$ m. The total system bandwidth and sampling rate are both $B=F_{s}=200$ MHz, enabling fine delay resolution. The OTFS lattice is defined as $Z_{M}\times Z_{N}$ with M=1024 delay bins and N=4096 Doppler/time bins. These parameters yield a subcarrier spacing of Δf=B/M=195.3125 kHz and an OTFS symbol duration $T_{\mathrm{sym}}=1/\Delta f=5.12\;\mu s$. The total frame duration is $T_{\mathrm{frame}}=N\times T_{\mathrm{sym}}=0.02097$ s, producing a Doppler spacing of $\Delta \nu =1/T_{\mathrm{frame}}=47.68$ Hz. The resulting delay resolution is $\Delta \tau =1/F_{s}=5$ ns (equivalent to 0.75 m in range), and the velocity resolution is $\Delta v=(\lambda_{r}/2)\Delta \nu \approx 0.76$ m/s. These values establish a high-fidelity sensing grid well-suited for maritime scenarios. All the simulations used the same lattice and these parameters. The full architecture of the project is shown in Figure 6, Table 1 summarizes the principal sea-state parameters for multiple frames.

The range of $H_{s}$ from 0.8–2.6 m reflects variation in overall wave energy, while $T_{p}$ decreases from 11.1 s to 4.6 s, marking the transition from long-period swells to short, choppy waves. The spectral shape factor γ varies from 2.0 to 6.6, indicating changes in spectral peakedness, and $U_{10}$ spans 6–11 m/s, representing moderate wind conditions. Each frame consistently includes 10 primary waves and 4 anomalies, ensuring balanced scene complexity. The dataset can be extended to Nth frames with Nth waves, so we have mentioned it for future work. Together with Figure 7 and Figure 8, these results confirm that the dataset encompasses a broad spectrum of marine environments suitable for validating OTFS-based sensing and learning frameworks. Each ocean component, positioned at range $r_{i}$ and moving with radial velocity $v_{r,i}$, contributes a propagation delay $\tau_{i}$ and Doppler shift $f_{d,i}$ defined as:(10)$$\tau_{i}=\frac{2r_{i}}{c},\;f_{d,i}=\frac{2v_{r,i}}{\lambda_{r}},\;\left|\alpha_{i}\right|\propto \frac{\sqrt{{\mathrm{RCS}}_{i}}}{r_{i}^{2}},$$
where $c=3\times 10^{8}$ m/s is the speed of light. The pair $\left(\tau_{i},f_{d,i}\right)$ determines the delay and Doppler coordinates of each reflection, which are then discretized into fractional OTFS bins $\left(m_{i}^{\left(f\right)},n_{i}^{\left(f\right)}\right)$. This mapping connects the physical ocean geometry directly to the radar’s delay–Doppler representation, forming the ground truth for both classical and AI-based sensing. Each anomaly produces a single complex reflection coefficient $\alpha_{i}$ that depends on its radar cross-section (RCS) and distance. The received amplitude decreases quadratically with range, while the RCS controls the overall signal strength. The magnitude of each tap follows:and the phase $∠\alpha_{i}$ is uniformly distributed in [−π,π). This relationship ensures realistic amplitude attenuation and random phase diversity, consistent with the physical radar scattering model.

Figure 7 illustrates a representative 1 km×1 km sea surface generated under JONSWAP statistics with $H_{s}$ m, $T_{p}$ s, γ, and $U_{10}$ m/s. The color scale encodes instantaneous surface elevation, highlighting the spatially varying energy distribution. The superimposed symbols denote the three anomaly types—rogue peaks (sharp crests), whitecaps (foam-topped waves), and floating debris (low-profile reflectors). These anomalies differ in both geometric height and effective RCS, resulting in distinctive delay–Doppler signatures. The figure confirms that the simulator reproduces realistic sea morphology and anomaly separability, forming a robust physical basis for subsequent OTFS sensing evaluation.

Figure 8 shows the spatial and statistical distribution of all simulated components across the dataset. Each point represents a wave or anomaly characterized by range $r_{i}$, azimuth $\theta_{i}$, velocity $v_{r,i}$, and reflection strength $|\alpha_{i}|$. The spread across both spatial and Doppler dimensions demonstrates broad coverage of marine dynamics, ensuring that the dataset spans calm to turbulent conditions. The anomalies occupy separable clusters in the delay–Doppler domain, validating that the generator preserves both spectral realism and class distinctness—critical properties for training AI-based radar sensing models.

### 4.4. Balanced and Oversampling

The learning set consists of complex, phase-normalized patches $x\in R^{N\times 2\times P\times P}$ (two real channels: in-phase and quadrature), paired with per-sample indices $\left(m,n\right)\in Z^{N}$ and regression labels $\left(\log\_\mathrm{amp},\mathrm{sinp},\mathrm{cosp}\right)\in R^{N}$. Optional aligned vectors include the frame ID fid, radial velocity $v_{r}$ [m/s], range *R* [m], and per-patch RMS. Additional bookkeeping keys (e.g., $\hat{m},\hat{n},\mathrm{wave}\_\mathrm{id}$) are preserved and passed through unchanged. Delay–Doppler data are inherently imbalanced: slow targets outnumber fast ones, near-range returns dominate far-range, and large-RCS anomalies are rarer than background waves. Training on such skewed data leads to biased models that overfit frequent regimes and underperform on rare but operationally important cases. Our balancing pipeline fixes this by (i) stratifying the dataset along physically meaningful axes (velocity, delay, amplitude/phase), (ii) oversampling under-represented strata to a user-specified target size, and (iii) adding phase-equivariant augmentation that respects the radar physics. The result is a training distribution that is both diverse and faithful to operational edge cases. Samples I={0,…,N−1} are partitioned into quantile bins along three axes: velocity $v_{r}$, delay index *m*, and amplitude/phase. Let $B_{V},B_{M},B_{A},B_{\phi }$ denote the number of bins for velocity, delay, log-amplitude, and phase, respectively. Bin edges are computed by empirical quantiles (with a uniform fallback for ties or missing values):(11)$$\begin{array}{cc} e_{i}^{\left(v\right)} & =\mathrm{Quantile}\left(v_{r},\;i/B_{V}\right),\;e_{i}^{\left(m\right)}=\mathrm{Quantile}\left(m,\;i/B_{M}\right), \end{array}$$(12)$$\begin{array}{cc} e_{i}^{\left(a\right)} & =\mathrm{Quantile}\left(\log\_\mathrm{amp},\;i/B_{A}\right),\;{\tilde{e}}_{j}^{\left(\phi \right)}=-\pi +j\;\frac{2\pi }{B_{\phi }}. \end{array}$$

Here, the instantaneous phase is represented continuously as (sinp,cosp), while ${\tilde{e}}^{\left(\phi \right)}$ provides a coarse partition for balancing. This stratification guarantees coverage across slow/fast, near/far, weak/strong, and different phase regimes. Given a total target count *T* per group (e.g., per axis) and a per-bin minimum $m_{0}$, we allocate samples to each nonempty bin *b* (with population $s_{b}$) either uniformly across nonempty bins or proportionally to their sizes. The policy is exposed via CLI flags, ensuring users can bias toward rare bins (uniform allocation) or preserve dataset shape (proportional). Sampling is with replacement inside bins to avoid exhausting rare strata. The union across axes (velocity *V*, delay *R*, and amplitude/phase *O*) forms the final index set $S=S_{V}\cup S_{R}\cup S_{O}$. Radar measurements are equivariant to global phase rotations. We exploit this by drawing *K* random phases $\theta_{k,u}∼U\left[-\pi ,\pi \right]$ for each selected index k∈S and rotating both the inputs and phase labels with the 2×2 rotation matrix(13)$$R\left(\theta \right)=\left[\begin{array}{cc} \cos\theta & -\sin\theta \\ \sin\theta & \cos\theta \end{array}\right].$$
Applied to the two input channels and to the label pair (cosp,sinp), this yields$$\left[\begin{array}{c} x_{ℜ}^{\prime }\\ x_{ℑ}^{\prime } \end{array}\right]=R\left(\theta \right)\;\left[\begin{array}{c} x_{ℜ}\\ x_{ℑ} \end{array}\right],\;\left[\begin{array}{c} {\mathrm{cosp}}^{\prime }\\ {\mathrm{sinp}}^{\prime } \end{array}\right]=R\left(\theta \right)\;\left[\begin{array}{c} \mathrm{cosp}\\ \mathrm{sinp} \end{array}\right].$$
Crucially, amplitude (log_amp) and physics descriptors ($v_{r},R$) are unchanged by global phase, preserving label correctness. This augmentation improves generalization without violating radar invariances. After augmentation, the raw size is $N_{\mathrm{raw}}=\left(K+1\right)\left|S\right|$. If a global cap is requested (-max-total$=N_{\max}$) and $N_{\mathrm{raw}}>N_{\max}$, we uniformly subsample to $N_{\max}$ and apply a joint shuffle over all arrays to preserve alignment of inputs, labels, and metadata. This guarantees reproducible runs and bounded memory. Quantile ties fall back to uniform bin edges on [min,max]; the last bin is right-inclusive to avoid boundary loss. Sampling is with replacement within a bin to ensure adequate representation of rare strata. If $v_{r}$ is unavailable, we skip velocity partitioning and balance over the remaining axes. All operations are stratified by split (train/val/test) to prevent leakage; the augmentation factor *K* is applied to train only. Binning is O(N), targeted sampling is linear in the selected counts, and the phase rotations cost $O(K\;S\;P^{2})$ for S=|S| patches of size P×P. In practice, with moderate *P* (e.g., P=11) and K∈[2,8], the pipeline is fast and easily GPU-parallelizable.

### 4.5. Fast Random-Forest Baseline

This baseline establishes a fast, interpretable machine-learning benchmark for the parameter-recovery task. Each complex patch $x\in R^{N\times 2\times P\times P}$ (real–imaginary channels) is associated with a discrete label y=m∈{0,…,M−1}, the true delay-bin index corresponding to its range cell. The objective is to predict *m* directly from the spatial–spectral texture of the patch. Every patch is flattened into a *d*-dimensional feature vector where $d=2P^{2}$. The full design matrix becomes(14)X=reshape(x,N×d).
Using a fixed random seed σ, a reproducible permutation $\pi_{\sigma }$ divides the data into training and validation subsets with $n_{\mathrm{tr}}=\left\lfloor \rho N\right\rfloor$ training samples and $n_{\mathrm{va}}=N-n_{\mathrm{tr}}$ validation samples, where ρ is the train fraction. This ensures deterministic benchmarking across runs. A Random-Forest classifier is trained with $n_{\mathrm{est}}$ trees, maximum depth $d_{\max}$, and minimum split size $n_{\mathrm{split}}$, using the Gini-impurity criterion:(15)$$\hat{h}\leftarrow \mathrm{RF}\left(n_{\mathrm{est}},d_{\max},n_{\mathrm{split}};\mathrm{seed}=\sigma \right),\;\hat{h}.\mathrm{fit}\left(X_{T},y_{T}\right).$$
Each tree $T_{j}$ produces a class vote, and the final decision is obtained by majority rule:$${\hat{y}}_{i}=\mathrm{mode}{\left\{T_{j}\left(X_{i}\right)\right\}}_{j=1}^{n_{\mathrm{est}}},\;i\in V.$$

This ensemble averaging suppresses overfitting and provides fast, non-parametric classification with complexity $O(n_{\mathrm{est}}\;n_{\mathrm{split}}\;n_{\mathrm{tr}}\log n_{\mathrm{tr}})$ during training and $O(n_{\mathrm{est}}\;\cdot \;\mathrm{depth})$ per-sample at inference. The RF model offers a physics-agnostic baseline that captures the statistical texture of amplitude–phase patterns in the delay–Doppler domain without relying on deep-network training or large GPUs. It is particularly useful for rapid diagnostics: by inspecting feature importance across flattened patch dimensions, one can identify which spatial–spectral regions contribute most to delay classification. Moreover, its deterministic random-seed initialization makes it ideal for reproducible benchmarking before introducing neural architectures. Performance is summarized using accuracy, macro-averaged precision, recall, and F1-score:$${\mathrm{Prec}}_{\mathrm{macro}}=\frac{1}{C}\sum_{c}\frac{{\mathrm{TP}}_{c}}{{\mathrm{TP}}_{c}+{\mathrm{FP}}_{c}},\;{\mathrm{Rec}}_{\mathrm{macro}}=\frac{1}{C}\sum_{c}\frac{{\mathrm{TP}}_{c}}{{\mathrm{TP}}_{c}+{\mathrm{FN}}_{c}},\;F1_{\mathrm{macro}}=\frac{1}{C}\sum_{c}\frac{2\;{\mathrm{Prec}}_{c}{\mathrm{Rec}}_{c}}{{\mathrm{Prec}}_{c}+{\mathrm{Rec}}_{c}}.$$
Here *C* is the number of classes (delay bins). These metrics ensure balanced evaluation across all range of categories, including rare far-range bins. We adopt a Random-Forest classifier on small, peak-centered, phase-normalized DD patches. This choice reflects our study constraints: moderate data scale, CPU-class millisecond latency, and the need for deterministic, reproducible behavior. Because upstream steps (oracle alignment, RMS scaling, reference-phase removal) already encode the core physics, RF provides a low-variance, fast, and stable decision rule without large training overhead or accelerator dependency. Higher-capacity deep models are not pursued here, as they would require expanded data, heavier tuning, and a distinct deployment pathway that is outside the present scope. Figure 9, Figure 10 and Figure 11 collectively evaluate the RF baseline trained on the balanced and phase-augmented dataset.

Figure 9 summarizes the macro-averaged metrics—accuracy, precision, recall, and F1-score—computed across all *C* delay classes m∈{0,…,M−1}, showing that the proposed quantile-bin balancing (velocity $B_{V}$, delay $B_{M}$, amplitude–phase $B_{A},B_{\phi }$) and targeted oversampling (*T*, $m_{0}$) significantly improve the recall and F1 of rare bins such as far-range or high-Doppler targets. The gains indicate that the model now generalizes more uniformly across the OTFS lattice, rather than favoring dense near-range regimes.

Figure 10 presents the multi-class Precision–Recall (PR) curves, where the macro Average Precision (AP) reaches 0.972, confirming excellent ranking of positives above negatives even when the curves exhibit small oscillations due to finite validation size. The high AP demonstrates that the classifier maintains robust ordering of correct classes across thresholds, validating that normalization (RMS, reference phase) and balanced sampling reduce bias from SNR variations.

Finally, Figure 11 displays the corresponding multi-class ROC curves, where most traces lie near the top-left corner, demonstrating strong separability of delay bins. Compared with an unbalanced baseline, the ROC curves shift upward after phase-equivariant augmentation (*K* random rotations), which enforces invariance to global carrier phase while exposing the model to diverse amplitude–phase orientations. Together, these figures confirm that the RF baseline trained on balanced and phase-augmented data achieves stable, physics-consistent performance, offering a reliable benchmark for subsequent deep-learning models.

## 5. Discussion

The results indicate that the proposed Marine OTFS–AI system’s quantitative performance aligns with practical sensing needs. The strong correlation between ground truth and the estimated range–velocity pairs confirms that the physical modeling is correct, while the AI refinement improves reliability under noise and multipath. Our evaluation includes quantitative metrics (RMSE, precision, recall, F1) and targeted figure interpretations. The AI module complements physics-based OTFS sensing by learning fine-grained amplitude–phase patterns in the DD grid. While the OTFS front end yields interpretable delay τ and Doppler $f_{D}$, slight inaccuracies from fractional bins or multipath can hinder direct analytical extraction. Trained on normalized complex patches $Y_{\mathrm{DD}}\;\in \;C^{M\times N}$, the model captures associations between DD intensity/phase curvature and the physical parameters (R,v). At inference, it consumes only received DD data—preventing any GT leakage—and maintains a pipeline suitable for dynamic marine environments.

Interpreting amplitude/phase vs. range/velocity: Root causes: (i) Per-frame complex ambiguity (CPE/gain) leaves (m,n) intact but rotates/scales taps; (ii) fractional delay–Doppler yields 2-D leakage, so the central-sample complex value depends on the window and nearby clutter; (iii) multipath superposition alters the complex coefficient more than the bin index; (iv) noise and phase wrapping are non-linear in angle. Implication: This does not indicate a fundamental limitation of OTFS sensing or of our pipeline; it reflects the identifiability of absolute complex coefficients under small residuals. Mitigations (analysis/calibration, no runtime bloat). (1) Pilot- or LS-based frame-wise complex-gain/CPE fit; (2) 2-D fractional-bin interpolation (e.g., quadratic/Quinn/Jacobsen-style) to reduce leakage bias; (3) window selection sweep (Hann/Kaiser) for leakage vs. resolution; (4) neighbor-aware debias in dense scenes; (5) robust reporting using 95th percentile amplitude ratio and circular phase error with SNR-aware pass bands. These steps raise complex-tap fidelity while preserving the deployed detector→patch→inference path.

Conventional ISAC sensing relies on (i) 2D-FFT with CFAR thresholding, valued for hardware efficiency but sensitive to leakage, near–far masking, and non-uniform clutter; and (ii) subspace/super-resolution estimators (e.g., MUSIC/ESPRIT), which can surpass grid resolution yet demand higher compute and robust covariance modeling. Our formulation differs by operating natively in the OTFS DD grid and confining computation to localized peak-centered patches with lightweight classification, thereby avoiding costly full-grid searches and retaining millisecond-class inference. In contrast, our OTFS CFAR → patch → RF pipeline remains near-linear in lattice size, requires no task-specific hardware IP, and achieves perfect bin-level localization (hence exact range/velocity) on held-out synthetic scenes. While we do not claim universal superiority over super-resolution methods in minimum-separation limits at very high SNR, our objective is complementary: physics-anchored, millisecond-class parameter extraction with explicit hooks for fractional/leakage calibration and a fairness protocol (SNR sweeps, close-spacing resolution curves, latency and footprint tables with confidence intervals).

(i) Sim → real shift can reduce accuracy; augmentation, domain adaptation, and few-shot fine-tuning help but may not remove all shift. (ii) Weak-label noise from CFAR/track pseudo-labels is mitigated by kinematic gating, temporal agreement, and robust losses. (iii) Amplitude/phase are more sensitive than delay/Doppler to global phase/CPE and fractional DD; frame-wise complex-gain calibration and phase-plane detrending provide auditing. (iv) Dense scenes with overlapping returns may require multi-frame association (JPDA/MHT) and/or higher pilot density. (v) Multi-frame logic is used during training/fine-tuning; runtime inference remains lightweight (ms-class). On constrained platforms, we preserve timing by (i) streaming rather than batch solves, (ii) avoiding full-grid global optimizations, and (iii) budgeting the number of peaks handed to the AI. If resources tighten, the system gracefully trades patch count for headroom without altering the detector’s core behavior. The result is a robust CPU-only implementation suitable for boats and buoys that value reliability, low power, and ease of maintenance.

RF’s capacity ceiling mainly affects continuous amplitude/phase fidelity, which is more sensitive to SNR and fractional leakage than discrete bin recovery. As we scale real-world scenes, lightweight deep heads (with pruning/quantization) can be investigated for amplitude/phase while preserving CPU-class latency; this is orthogonal to the present contribution focused on deterministic oracle labeling and interpretable, low-variance inference. We will validate the system in X-band (9.4 GHz) with the same OTFS lattice (M = 1024, N = 4096, CP = 1024, $B=F_{s}=200$ MHz) to preserve one-to-one lattice mapping. Data will be collected across harbor, coastal, and offshore sites spanning Douglas 1–5. Ground truth will combine wave buoys (Hs, Tp), fixed corner reflectors (range anchors), Automatic Identification System (AIS) vessel tracks, and Global Navigation Satellite System, Inertial Measurement Unit (GNSS/IMU) tags on a maneuvering workboat; shore Uncrewed Aerial Vehicles cameras/UAVs provide qualitative cross-checks. Runtime remains detector→patch→inference (ms-class). Optional fine-tuning uses weak/self-supervision: track-gated pseudo-labels (Kalman/IMM; optional JPDA/MHT), Mahalanobis/innovation gating, temporal stability over *K* frames, confidence weighting, and early stopping on a small vetted subset.

## 6. Conclusions

We presented a physics-anchored OTFS sensing pipeline with a lightweight learning head that operates on peak-centered delay–Doppler patches. Using oracle association only in simulation for unambiguous labels—and a GT-free detector → patch → inference path at deployment—the system achieves millisecond-class runtime in our experiments on a CPU-centric setup while preserving interpretability. On a balanced 10k corpus (80/20 split), the learned head reached 0.966 accuracy, 0.965 macro-F1, and 0.998 macro ROC–AUC. In held-out scenes, discrete delay/Doppler bins—and hence range and radial velocity—matched ground truth at 100%, with 89% agreement for amplitude/phase. These results confirm exact lattice mapping and reliable recovery of the primary sensing KPIs (range/velocity) at low computational cost. Limitations primarily affect continuous amplitude/phase due to residual CPE, fractional-bin leakage, and multipath. We treat these as secondary diagnostics and, when needed, apply analysis-only calibrations (frame-wise complex-gain fits, phase-plane detrending, fractional-bin interpolation) without altering the deployed runtime. Planned work focuses on field validation in X-band (9.4 GHz) with the same OTFS lattice across Douglas 1–5 sea states using multi-source ground truth, alongside weak/self-supervised fine-tuning (track-gated pseudo-labels with Kalman/IMM; optional JPDA/MHT). Overall, the OTFS CFAR → patch → RF design delivers physics-consistent, deployable sensing with exact bin recovery and predictable latency, providing a strong baseline for marine ISAC applications.

## Figures and Tables

**Figure 1 sensors-25-07104-f001:**
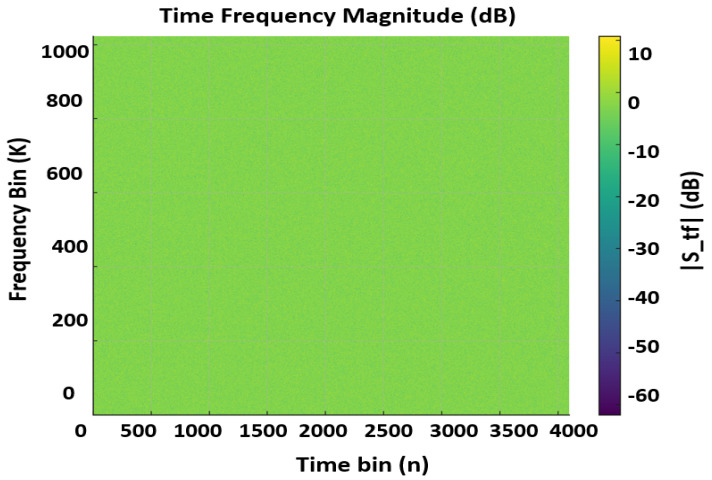
Wave visualization of OTFS signal in TF (dB) Domain.

**Figure 2 sensors-25-07104-f002:**
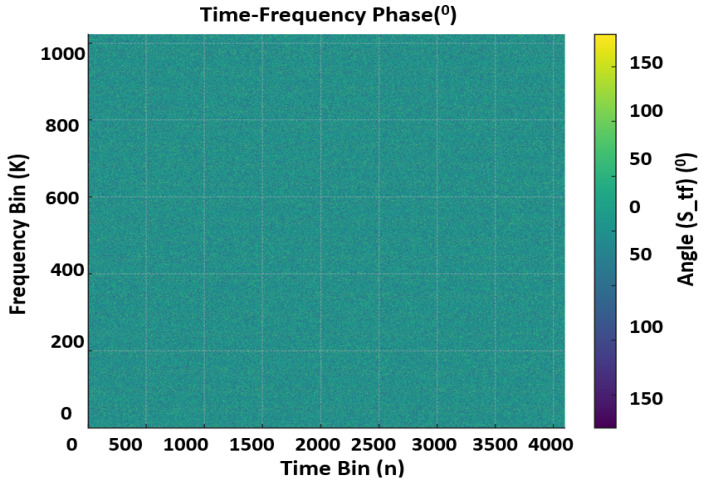
Wave visualization of OTFS signal in TF (degree).

**Figure 3 sensors-25-07104-f003:**
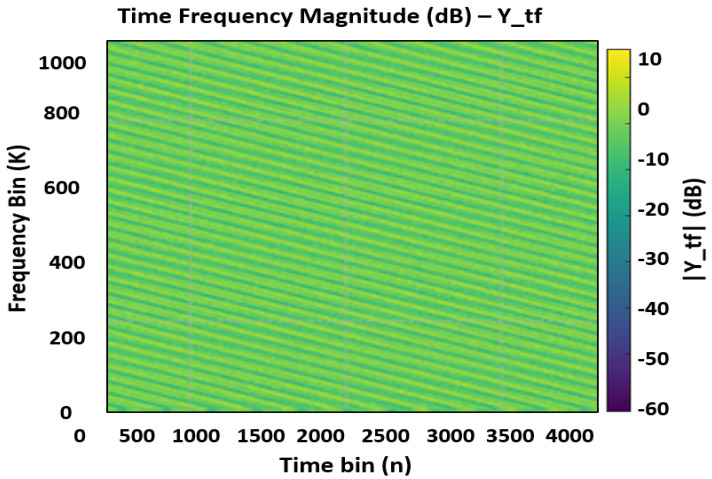
Noisy Rx Signal in TF domain (dB).

**Figure 4 sensors-25-07104-f004:**
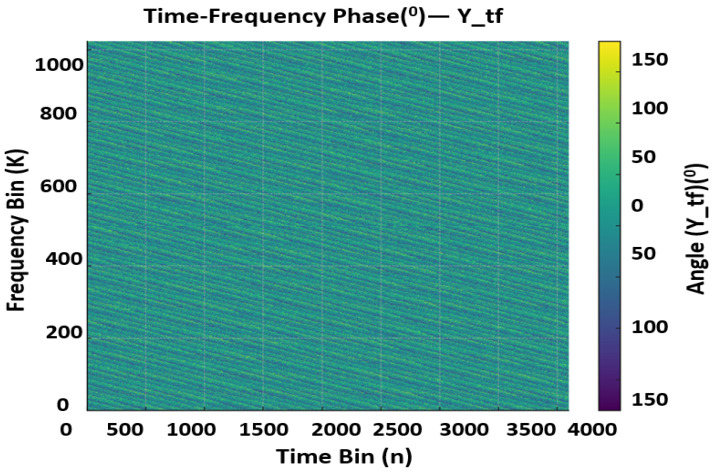
Noisy Rx Signal in TF domain (degree).

**Figure 5 sensors-25-07104-f005:**
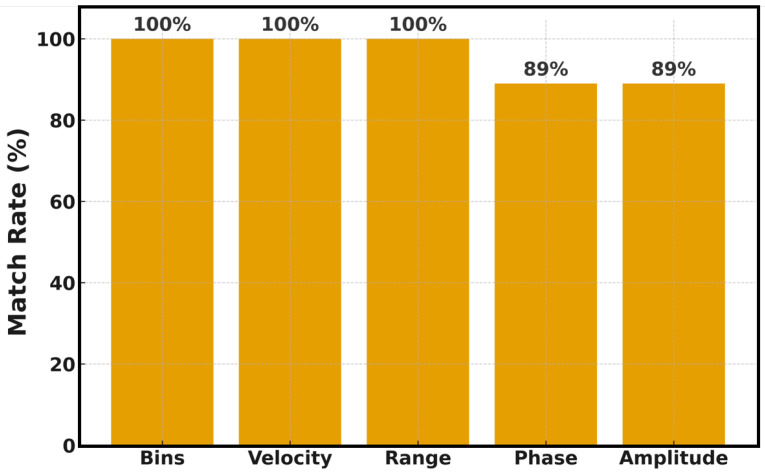
Match rate of different parameters.

**Figure 6 sensors-25-07104-f006:**
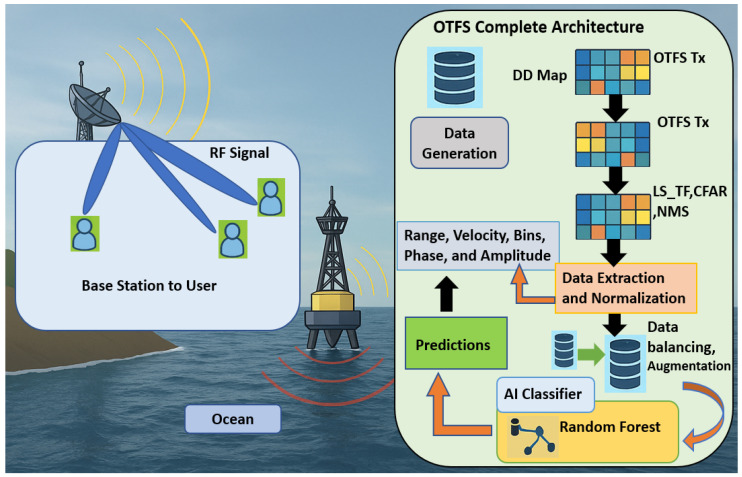
Full end-to-end view of sensing and AI.

**Figure 7 sensors-25-07104-f007:**
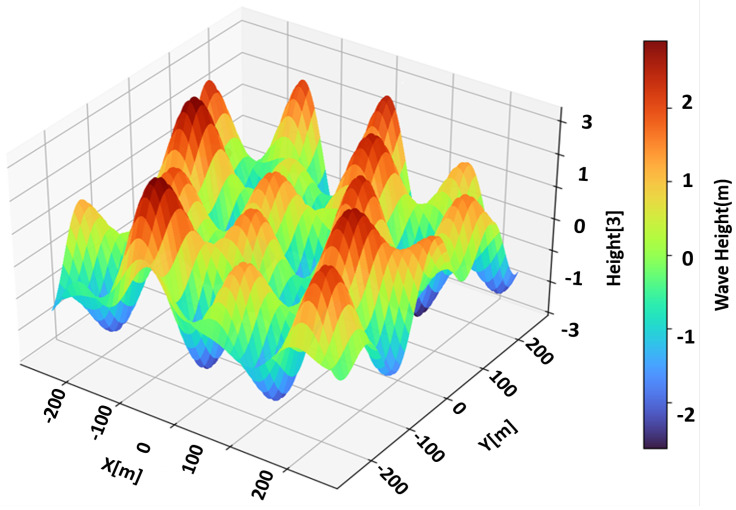
Visualization of a generated marine scene.

**Figure 8 sensors-25-07104-f008:**
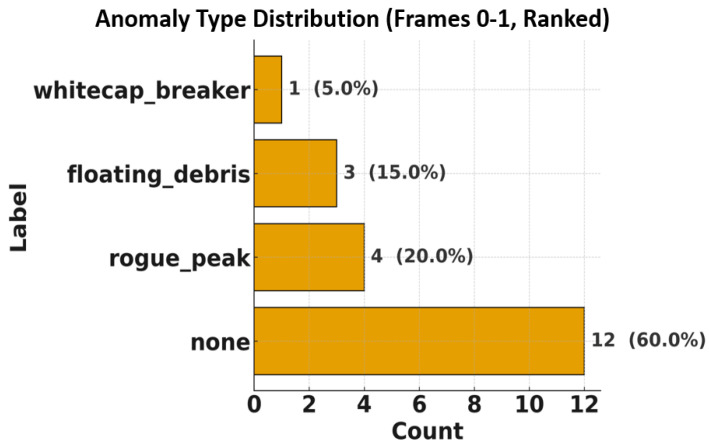
Distribution of waves and anomalies in the dataset.

**Figure 9 sensors-25-07104-f009:**
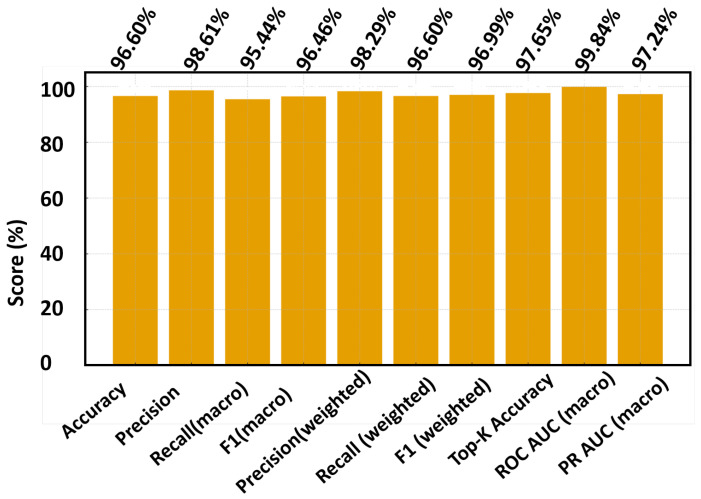
Macro-averaged performance of the RF baseline after balancing and augmentation.

**Figure 10 sensors-25-07104-f010:**
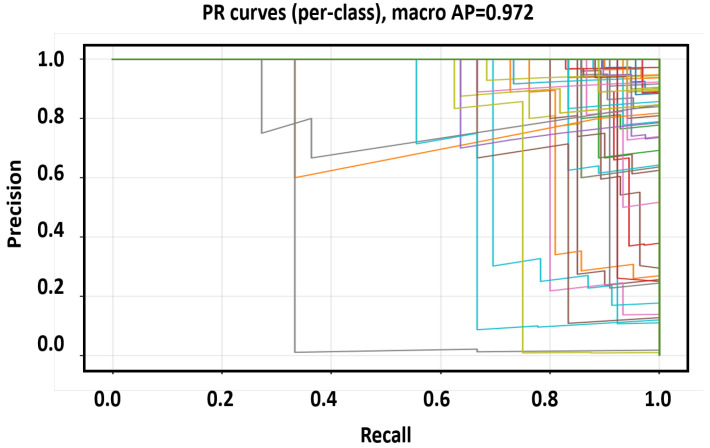
Multi-class Precision–Recall (PR) curves of the RF baseline.

**Figure 11 sensors-25-07104-f011:**
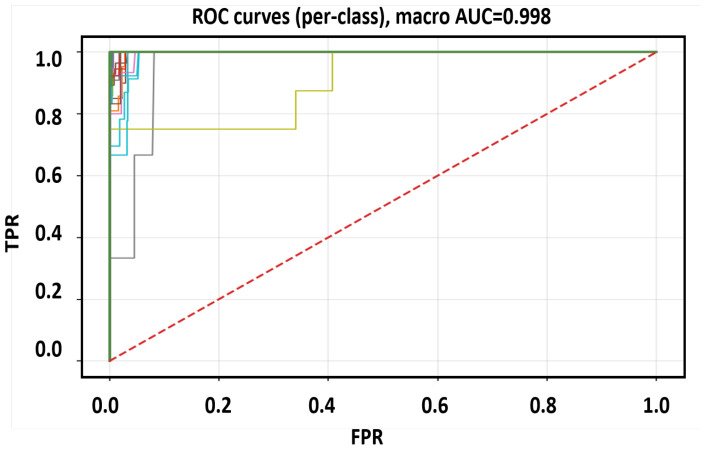
Multi-class ROC curves of the RF baseline showing consistent separability across range classes.

**Table 1 sensors-25-07104-t001:** Per-frame sea-state summary with wave/anomaly counts. $U_{10}$ denotes wind speed at 10 m.

Frame	$H_{s}$ [m]	$T_{p}$ [s]	γ	$U_{10}$ [m/s]	Waves/Anoms
0	0.84	11.10	6.60	9.46	10/4
1	2.17	9.76	5.70	8.21	10/4
⋮	⋮	⋮	⋮	⋮	⋮
9	2.62	4.66	2.05	6.10	10/4

## Data Availability

All the data are available in the paper. The code and other related data will be provided on request.

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
