# Peer review of "Low-Latency Marine-Based OTFS Echo Parameter Estimation Enabled by AI"

_sensors, 2025, doi:10.3390/s25237104_

Round 1
Reviewer 1 Report
Comments and Suggestions for Authors
This paper presents a hybrid OTFS-based sensing framework that integrates deterministic signal processing with a machine learning stage, specifically using a Random Forest classifier for parameter estimation in marine environments. The reviewer has the following questions:
1. How does the oracle ground-truth association process handle scenarios where exact ground-truth labels are unavailable in real-world deployments, and what are its limitations?
2. Given the reliance on simulated data, what steps are planned to validate the model's performance in actual marine conditions with unpredictable noise and clutter?
3. Why does the amplitude and phase recovery achieve only 89% match compared to the 100% success in range and velocity, and does this indicate inherent constraints in the method?
4. How does the proposed method compare quantitatively against traditional techniques like MUSIC or FFT-based approaches in terms of computational complexity and accuracy under similar conditions?
5. What motivated the selection of Random Forest over other AI models, and were alternatives like deep neural networks evaluated for potential gains?
6. Could the author elaborate on the hardware requirements and latency benchmarks for real-time implementation, especially for resource-constrained marine platforms?
Reviewer 2 Report
Comments and Suggestions for Authors
Paper title: Low-Latency Marine-Based OTFS Echo Parameter Estimation
Enabled by AI
Authors propose an end-to-end pipeline for Orthogonal Time-Frequency Space (OTFS) sensing that integrates deterministic signal processing with a Machine Learning (ML) inference stage. The pipeline first generates a complex delay–Doppler grid via standard Symplectic Fast Fourier Transform (SFFT)-based OTFS reception. Authors employ an ’oracle’ Ground-Truth (GT) association process to deterministically label signal peaks, extracting their complex gain (α) and absolute indices (m, n) to deduce physical targets (range, radial velocity). These oracle-aligned labels are used to train a Random Forest (RF) classifier. The RF model learns to map normalized 33 × 33 complex patches, centered on signal peaks, to their corresponding target parameters.
I have some questions:
- Authors should include a system architecture figure and AI applications in this figure. It will be good for readers.
- Also, authors should insert the system architecturte section in Section 2.
- Section 2: data preprocessing should be put in section 5: AI implementation. Before putting data in the AI model, we need to preprocess data. It will be kept consistent.
- In section 5, I see some results; however, I did not see some setup parameters: OFDM length, cyclic-prefix length, Doppler shift, etc. If possible, authors should make a result table with different OFDM lengths, Doppler shifts, etc., and the performance of AI models. It is good to highlight your works.
- The conclusion part is a bit short; the authors should write more.
In my point of view, this paper should have minor revision.
Round 2
Reviewer 1 Report
Comments and Suggestions for Authors
I have no other questions.